# Automatically delineating the calving front of Jakobshavn Isbræ from multi-temporal TerraSAR-X images: a deep learning approach

Enze Zhang[1], Lin Liu[1], Lingcao Huang[1]

[1]Earth System Science Programme, Faculty of Science, The Chinese University of Hong Kong, Hong Kong, China.

*Corresponding to*: Enze Zhang (zhangenze@link.cuhk.edu.hk)

**Abstract.** The calving fronts of many tidewater glaciers in Greenland have been undergoing strong seasonal and inter-annual fluctuations. Conventionally, calving front positions have been manually delineated from remote sensing images. But manual practices can be labor-intensive and time-consuming, particularly when processing a large number of images taken over decades and covering large areas with many glaciers, such as Greenland. Applying U-Net, a deep learning architecture, to multi-temporal Synthetic Aperture Radar images taken by the TerraSAR-X satellite, we here automatically delineate the calving front positions of Jakobshavn Isbræ from 2009 to 2015. Our results are consistent with the manually delineated products generated by the Greenland Ice Sheet Climate Change Initiative project. We show that the calving fronts of Jakobshavn's two main branches retreated at mean rates of -117 ± 1 m yr$^{-1}$ and -157 ± 1 m yr$^{-1}$, respectively, during the years 2009 to 2015. The inter-annual calving front variations can be roughly divided into three phases for both branches. The retreat rates of the two branches tripled and doubled, respectively, from phase 1 (April 2009–January 2011) to phase 2 (January 2011–January 2013), then stabilized to nearly zero in phase 3 (January 2013–December 2015). We suggest that the retreat of the calving front into an overdeepened basin whose bed is retrograde may have accelerated the retreat after 2011, while the inland-uphill bed slope behind the bottom of the overdeepened basin has prevented the glacier from retreating further after 2012. Demonstrating through this successful case study on Jakobshavn Isbræ and due to the transferable nature of deep learning, our methodology can be applied to many other tidewater glaciers both in Greenland and elsewhere in the world, using multi-temporal and multi-sensor remote sensing imagery.

## 1 Introduction

Glacier retreating is one of the processes that control the recent speedups of Greenland's tidewater glaciers (King et al., 2018). As glacier retreats, it accelerates to compensate for the loss of downstream buttress. Glacier dynamic instabilities, as suggested decades ago by Meier and Post (1987), play an essential role as the glaciers retreat over depressions in the bedrock topography. For example, Joughin et al. (2008a) indicated that dynamic instabilities caused Helheim and Kangerdlugssuaq Glaciers to speed up as they retreated into an overdeepened basin whose bed is retrograde between 2001 and 2006. Examining 276 marine-terminating outlet glaciers, Bunce et al. (2018) concluded that bed geometry is an important control on the timing and magnitude of glacier retreat.

An accurate and detailed quantification of calving front variations would improve our understanding of the controlling mechanisms of glacier retreat. Moreover, observations of retreat may serve as initial indicators for other dynamic variations such as the glacier acceleration (Moon and Joughin, 2008). Calving front positions are influenced by a range of forces including ice mélange buttressing, increased runoff, and ocean-driven melt (Moon et al., 2015; Fried et al., 2018). Nevertheless, the mechanisms behind the numerous and complex controls on front positions are not yet fully understood.

Compared with manually digitizing calving fronts, automatic mapping is superior because of greater productivity and reliability and lower cost. While most of the previous studies have manually delineated the calving fronts (e.g., Howat et al., 2005; Joughin et al., 2008b), studies by Sohn et al. (1996) and Seale et al. (2011) have automatically delineated calving fronts using feature extractors. Sohn et al. (1996) designed a method to extract ice sheet margin by applying Roberts edge extractor to ERS-1 Synthetic Aperture Radar (SAR) images. Seale et al. (2011) automatically identified glacier calving fronts from daily MODIS images by combining Sobel and brightness profiling methods. With low computational complexity requiring no training and little memory resources, these feature-extracting methods are promising but require extensive prior knowledge and experience. Deep learning method has also been applied to delineate the calving front positions. Mohajerani et al. (2019) have applied U-Net architecture to Landsat-5, -7, and -8 images over Jakobshavn, Sverdrup, Kangerlussuaq, and Helheim glaciers.

Deep learning can solve more complex problems with little prior knowledge required and take advantage of increased data volume (LeCun et al., 2015). With the continuous accumulation in the past decades and in recent space missions, the data volume of remote sensing imagery in the polar regions has increased dramatically. Moreover, glacier systems are complex, as conditions such as weather and glacier dynamic behaviors vary from place to place and from season to season. There are therefore obvious advantages to applying deep learning techniques to automatically extract glaciological features from the available big data.

Here we aim to design a method to automatically delineate a glacier calving front from multi-temporal TerraSAR-X (TSX) images based on deep convolution neural networks (DCNNs). More specifically, we delineate the glacier calving front of Jakobshavn Isbræ (Fig. 1a) and quantify its seasonal and inter-annual variations. With this new set of observations, we investigate the possible link between calving front variations and bed elevation.

DCNNs are a class of the deep learning methods, and have made important breakthroughs in image processing. DCNNs can discover both low-level (e.g., edges, corners, and lines) and mid-level features (e.g., shapes, sizes, and locations) (Sun et al., 2014; Zhang et al., 2015). Recently, some studies have used DCNNs on high-resolution SAR images to perform classification tasks (Geng et al., 2015; Huang et al., 2017). These studies unanimously agree that DCNNs outperform traditional classification methods on SAR images.

We use TSX images due to their high temporal resolution (11 days), high spatial resolution (3.3 to 3.5 meters), and ability to penetrate cloud cover. These high-temporal-resolution images have been acquired in all seasons and allow us to investigate calving front variations with a high degree of continuity and consistency. With these high-spatial-resolution images, we can easily digitize the calving fronts (known as "ground truth" in the context of deep learning), and verify the accuracy of the

DCNN. Using SAR images can avoid the cloud cover problem associated with optical images such as the Landsat-8 image shown in Fig. 1b.

## 2 Jakobshavn Isbræ

Jakobshavn Isbræ, located in central-west Greenland, is one of the largest and fastest tidewater glaciers in the world. In Jakobshavn, the ice flows westward to the ocean and divides into two branches near the coast (Fig. 1a). Before summer 2004, these two branches merged and flowed into the Kangia fjord. Afterwards, as the glacier retreated, the two branches became disconnected (Bondzio, 2017). During the past few years, Jakobshavn Isbræ has undergone dramatic acceleration as the glacier has retreated and thinned (Joughin et al., 2008c; Joughin et al., 2012). Jakobshavn's calving front retreated 16 km between 2002 and 2008 (Rosenau et al., 2013). This glacier alone has contributed nearly 1 mm to the global sea level rise from 2000 to 2011 (Howat et al., 2011).

Observations have shown that the calving front variations were correlated with the glacier velocity changes in Jakobshavn Isbræ. In 1998, the glacier sped up by 18% in its frontal regions, coinciding with the initial retreat of the ice tongue (Thomas, 2004; Luckman and Murray, 2005). The glacier doubled its speed by spring 2003, when nearly the entire floating ice tongue had disintegrated (Joughin et al., 2004). After the loss of this ice tongue, the glacier's velocity fluctuated seasonally from 2004 to 2007 (Joughin et al., 2008b). The glacier slowed down when it was advancing, and sped up when it was retreating (Joughin et al., 2012).

The variations of Jakobshavn's calving front are also strongly influenced by the presence of ice mélange, namely a mixture of calved icebergs and sea ice (Fig. 1a). The seasonal variation of the calving front in Jakobshavn Isbræ is well correlated with the growth and recession of sea ice in the Kangia fjord (Sohn et al., 1998; Joughin et al., 2008c). Temporal variations of the ice mélange strength can also control the timing of calving events and influence the evolution of the calving front position (Amundson et al., 2010).

Our study area covers a 14×18 km section of the frontal area of Jakobshavn, and includes bedrock, ice mélange, and glacier regions. We restrict the extent of the study area to reduce the computational costs, while also ensuring the coverage of all the calving fronts within our investigation period (2009–2015, determined by the TSX images we have access to). We classify our study area into two classes: ice mélange and non-ice mélange regions (including both glacier and bedrock regions). We delineate the boundaries between these two regions and retrieve glacier calving fronts. The repetitive texture of crevasses in the glacier region clearly distinguishes it from ice mélange, where icebergs are distributed discretely. It is easy to identify the bedrock region because of the distinct bedrock texture, including cracks and land-based lakes.

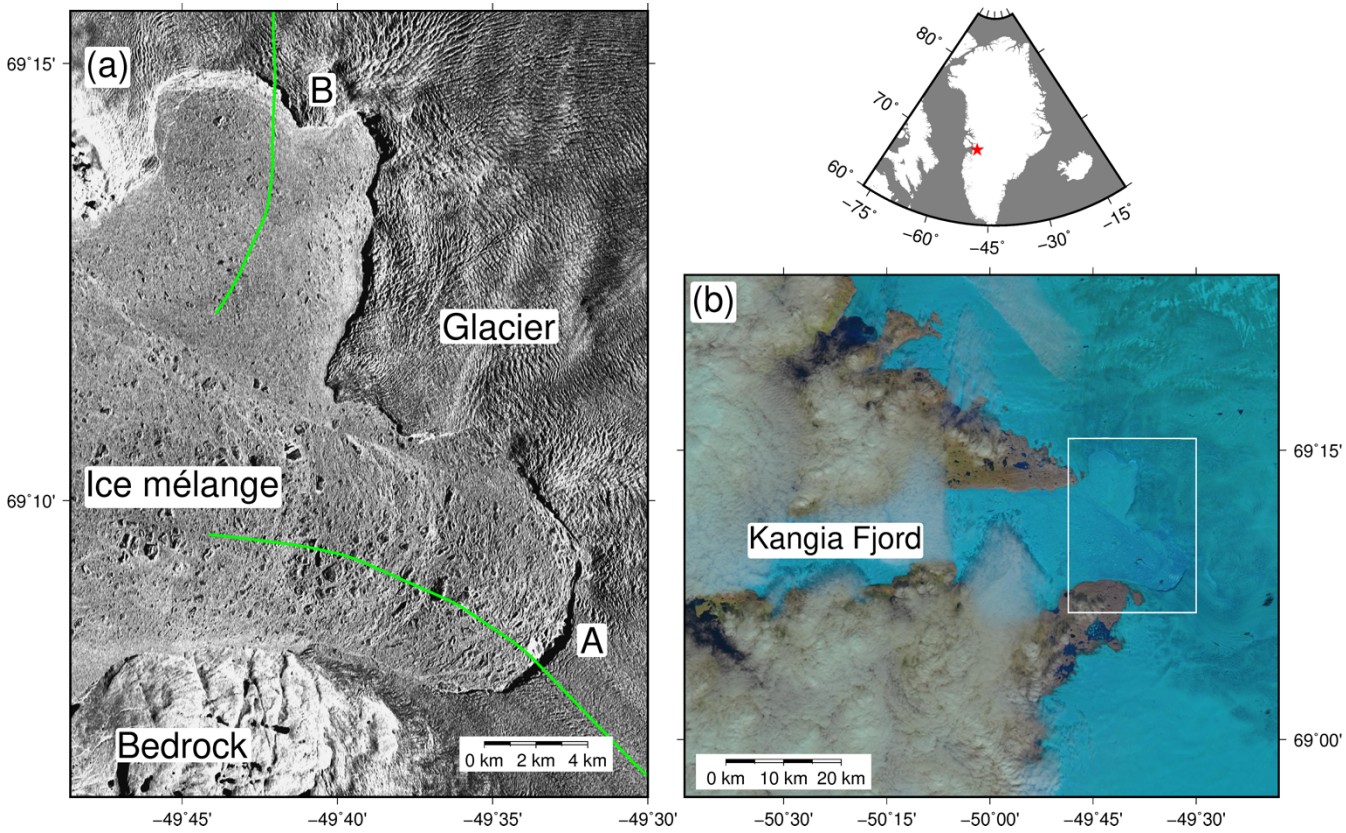

**Figure 1.** (a) TerraSAR-X image taken on 11ᵗʰ July 2015 showing the frontal area of Jakobshavn Isbræ. Its two branches are labeled as 'A' and 'B'. The green lines indicate the location of the bed elevation profiles shown in Fig. 8. (b) Landsat-8 image taken on 13ᵗʰ July 2015. The white box shows the area illustrated in Fig. 1a.

## 3 TerraSAR-X images and pre-processing

The German SAR satellite TerraSAR-X was launched in June 2007 and carries an X-band SAR sensor. In this study, we use TSX images taken in both ascending and descending orbits and in stripmap imaging mode. We use the enhanced ellipsoid corrected (EEC) products, which are multi-looked, projected and resampled to the WGS84 reference ellipsoid. We use 159 images in total, taken between April 16ᵗʰ, 2009 and December 23ʳᵈ, 2015 (listed in Table S1). We apply three pre-processing procedures including despeckling, multi-looking, and re-georeferencing. Figure 2 shows an illustrative example of our pre-processing workflow, which we will describe below in detail.

Because the quality of SAR images is adversely affected by the speckle noise (Fig. 2a), we apply the median blur filter to mitigate the speckle noise (Fig. 2b) and then multi-look the filtered images to reduce their size by 25 times (Fig. 2c). The median blur filter is widely used in image processing and is particularly effective for speckle noise. With the despeckled images, we average five neighboring pixels (vertically and horizontally) by using Geospatial Data Abstraction Library (GDAL) package ([www.gdal.org](www.gdal.org)). Moreover, both despeckling and multi-looking can smooth images without the loss of

essential information for delineating the calving fronts. After despeckling and multi-looking, the pixel size of our images is six meters.

We choose the EEC products because they include topographic correction and are the standard geocoded products of TSX (Roth et al., 2004). However, even for the EEC products, we observe that the geocoding information for our study area is inaccurate. First, overlaying the EEC images on Google Earth, we note obvious offsets between these two. Second, the geocoding information is inconsistent in different orbit directions of EEC products. Therefore, we need to re-georeference the EEC products. For the images we have, we observe that the images in the same orbit direction have identical geometry. Based on this observation, we assume that the differences between the EEC products and the Google Earth images are systematic, namely that they are consistent for the EEC products in the same orbit direction. We correct the geocoding information of the EEC products using 16 ground control points on Google Earth images, including the center of lakes and cross sections of the bedrock, due to their stability and ease of identification. For all the EEC products in the same orbit direction, we apply the same thin plate spline transformation using the GDAL package.

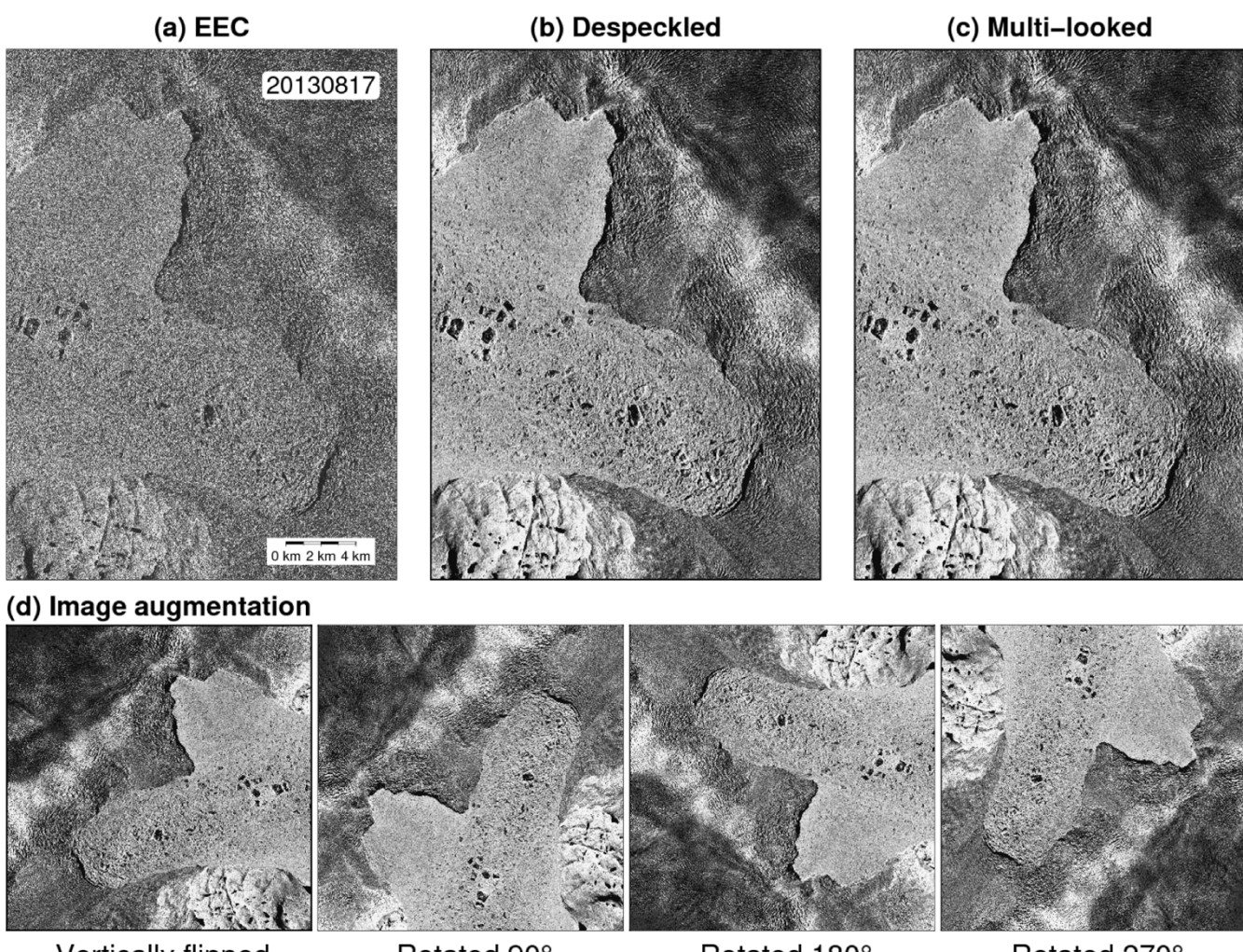

**Figure 2. A set of examples of TerraSAR-X data pre-processing and preparation, including (a) the enhanced ellipsoid corrected (EEC) product, (b) despeckled image after reducing the speckle noise, (c) multi-looked image after decreasing the image size, and (d) images after vertically flipping and rotating Fig. 2c by 90º, 180º, and 270º, respectively. For ease of presentation, the images in Fig. 2d are not to scale with (a)–(c).**

## 4 Deep learning and post-processing

DCNNs are a class of neural networks that consist of numerous convolutional layers, each of which contains learnable weights and biases. A network's architecture refers to its overall structure, including the number of units and layers the network has and how they are connected. Here, we use the U-net architecture, which has achieved outstanding performance in biomedical segmentation applications and is among the best methods in image segmentation (Ronneberger et al., 2015). This network is fast, taking less than a second on a mainstream graphics processing unit (GPU) to segment a 512×512 image. The U-net architecture consists of a contracting path and an expansive path (Fig. S1). The contracting path consists of

repeated application of two 5×5 convolution layers, each followed by a batch normalization layer and a leaky rectified linear unit (LeakyReLU) activation function, and 2×2 max pooling operation for downsampling feature maps and doubling the number of feature channels. Every step in the expansive path consists of a 4×4 up-convolution layer that upsamples the feature map and halves the number of feature channels, a concatenation with the corresponding feature map from the contracting path and two 5×5 convolution layers, each followed by a batch normalization layer and a LeakyReLU activation function. The final layer is a 3×3 convolutional layer with Sigmoid activation function to get the final segmentation patch. We utilize relatively large convolution kernel size (5 by 5) to obtain smoother calving fronts. We use LeakyReLU activation functions with a slope of 0.1 below zero, which allows for small, non-zero gradient when the unit is not active (Mass et al., 2013), making optimization potentially more robust. We use binary cross-entropy (BCE) between the ground truth images, and the network outputs to measure the training error because it avoids the problem of slow learning (the training loss decreases slowly) (Goodfellow et al., 2016). We use adaptive moment estimation max (AdaMax) (Kingma and Ba, 2014) as the optimizer with a learning rate of 0.0001 and an L2 regularization factor of 0.00001.

The proposed framework for using deep learning to delineate the calving fronts is summarized in Fig. 3. We separate all the SAR images into a training-validation dataset (75 images) and a test dataset (84 images) (Table S1). In the training-validation dataset, we randomly choose 90% as training data and take the rest as validation data. The validation dataset is for minimizing overfitting and tuning the hyperparameters of the network such as learning rate and kernel size.

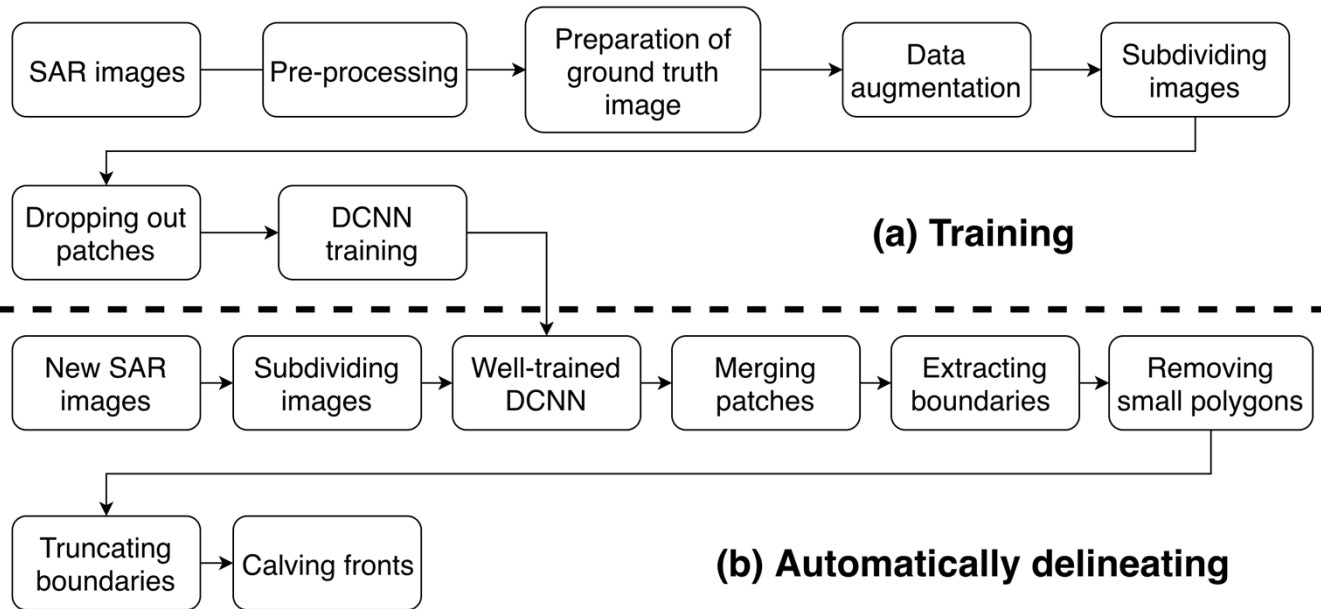

**Figure 3. Diagram of the proposed framework. Details are described in Sections 3 and 4.**

Before training the network, we prepare the training dataset, including training images (SAR images) and their corresponding ground truth images. The ground truth images have two classes: the ice mélange region is set as zero, and the non-ice-mélange region (including both glacier and bedrock regions) is set as one (Fig. S2). The ground truth images are

derived by converting the vector of manually delineated calving fronts to rasters using GDAL. Manual delineation is simple on most TSX images. However, it is challenging to delineate the calving fronts on few TSX images acquired in winter and spring because the boundaries are obscured due to snow cover and sea ice bonding. For each of these obscure images, we use its temporally closest image with a clear calving front as a reference, and require that our manually-delineated fronts are smooth (Fig. S3).

To ensure the effectiveness and accuracy of deep learning for a set of SAR images taken in all seasons spanning seven years, it is important to prepare a sufficiently diverse training dataset. We include at least one image in each month into the training dataset to represent various conditions related to radar backscatter and image texture. First, radar backscatter can vary with the dielectric properties of the surface scatterers in the study area due to changes in snow cover, wetness, and variations in geometric properties such as roughness, grain size, and internal structure (Fahnestock et al., 1993). Since our study area is in the ablation region, backscatter increases in winter because of dry snow cover and decreases in summer due to snow melting. Second, the seasonal and inter-annual variations of ice mélange condition can change the image texture. Sea ice formation in winter solidifies ice mélange, while ice mélange weakens in summer, resulting in freely floating icebergs (Amundson et al., 2010; Xie et al., 2016).

We also perform data augmentation to enrich our training dataset. We adopt the following two strategies. First, we vertically flip and rotate our training images by 90°, 180°, and 270°, respectively, to constitute many possible locations of the calving front in the study area (Fig. 2d). Second, we apply 2% linear stretch to the training images to enhance the edges. For all the values between the $2^{nd}$ and $98^{th}$ percentiles of the pixel value histogram, we linearly stretch them to the range between 0 to 255. The values lower than the $2^{nd}$ percentile are set to zero, and the values larger than the $98^{th}$ percentile are set to 255.

We subdivide each image (3565×1634 pixels) in the training dataset into small patches (960×720 pixels). Otherwise, the resolution would be limited by the GPU memory. We split images with overlaps, and obtain 36414 patches in total. Such a strategy allows a seamless segmentation after merging, which reduces the edge effect. A larger patch size can also better mitigate the edge effect. A common training strategy in deep learning is to train several training examples as a batch each time instead of training the whole dataset. With a given GPU memory, a smaller patch size allows more items in a batch, which increases the efficiency and improves the accuracy of the gradient estimation at each step. To strike a balance between edge effect and batch size, we choose 960×720 pixels as our patch size and the batch size is three.

Due to different computational time used in training and automatic delineation, the overlap areas between adjacent patches are set differently in the training and the test datasets. Taking the GPU we use as an example, training the network takes more time (80 hours) than automatic delineation (20 minutes) after the network is well trained. Therefore, we split the training images with smaller overlap (two-thirds of the patch size) to save computational power and split the test images with larger overlap (four-fifths of the patch size) to make denser samplings so that the results become more robust.

Balancing the number of training samples between classes is crucial in deep learning (Batista et al., 2005; Anantrasirichai et al., 2018). Compared with patches with two classes, patches with only one class are not equally helpful for delineating

boundaries. However, one-fifth of the 36414 patches only have one class. Therefore, we randomly drop out 80% of these patches to make the network perform better on the boundary between two classes and also to save computational power.

Training the network starts with initializing all weights as zero. We stop the training when the validation error starts to increase for five consecutive epochs. After the training, we first subdivide each test TSX image into small patches and use the well-trained network to segment all the patches into ice mélange and non-ice-mélange classes. Then, we merge the segmented patches (binary images with a pixel value of one or zero) into a single segmentation image by averaging the overlaps. After merging, if the pixel value is larger than 0.5, we consider the pixel to be in a non-ice mélange region. We use GDAL to convert the segmentation image into a vector, which contains a large polygon constituted by both the calving front and the image border, and small isolated polygons caused by erroneous segmentation. After removing the small polygons and truncating the large polygon to separate the calving front from the image border, we finally obtain the calving front for each image.

Using the post-processed delineation results, we can quantify the temporal calving front variations of both branches. Taking the earliest calving front (April 16th, 2009) as the reference, we calculate the enclosed area bounded by the reference and the calving front in a given TSX image. We adopt these metrics of area changes because they take both calving front position and shape into account.

## 5 Data validation and error estimation

Our results are validated by calving front products from the Greenland Ice Sheet Climate Change Initiative (CCI) project (http://products.esa-icesheets-cci.org). The CCI calving fronts are derived by manual delineation using ERS & Sentinel-1 SAR, and Landsat-5,7,8 optical imagery. We validate our results in the following two aspects.

First, the validation of the re-georeferencing (Section 3) is derived by directly comparing the manually delineated calving fronts obtained from this study and the CCI products. The calving fronts from these two datasets should be on the same date, and therefore, only six calving fronts are compared. We manually delineate the calving fronts from the TSX image after re-georeferencing and then calculate the averaged width of the enclosed area bounded by both the calving fronts from these two datasets. The mean difference is 104 meters (equivalent to ~17.3 pixels) (Table S2 and Fig. S4). Several reasons could cause such a seemingly large difference. The geocoding information of the CCI products also has uncertainties. Moreover, manual delineation from both the CCI and ours are subjected to image quality and the different criteria we adopt for front delineation. To measure the manual delineation error, we have another investigator to manually delineate the above-mentioned six calving fronts again. By comparing the two sets of independent delineation results, we obtained a mean difference of 33 meters (equivalent to ~5.5 pixels) (Table S2).

Second, the difference of calving front variations between ours and the CCI presents an overall validation that sums up both re-georeferencing and network-delineation uncertainties. We quantify the calving front variations of the CCI products with

the same method and reference used in our results. Finally, we calculate the difference between these two variations in terms of both area and equivalent length.

The errors in the test dataset represent the error of the network. Unlike the BCE-measured segmentation error in training, the test error is for calving front delineation. We measure the test error by calculating the averaged width of the enclosed area bounded by the manually delineated and the network-delineated calving fronts (Fig. S5).

## 6 Results

We present our results in the following order: (1) the network-delineated calving fronts from April 16th, 2009 to December 23rd, 2015, which are shown in a movie (Movie S1); (2) two examples of our automatically delineated calving fronts (i.e., results in test dataset) (Fig. 4); (3) retreat rates (Table 1) and time series of calving front variations (Fig. 5); (4) inter-annual calving front variation (Fig. 6 and 7).

The individual network-delineated results are influenced by image quality. Usually, the boundary is more distinct in summer than in other seasons, yielding superior results (Fig. 4a). In winter and spring, the boundary is obscure due to the low contrast and similar texture of the images, for example, the Branch B and the northern part of Branch A (Fig. 4b, 4c, and 4d). The backscatters of the snow-covered ice mélange and the glacier are similar. Moreover, sea ice formation in winter solidifies the ice mélange and even bonds it with the glacier. As a result, our detected edge deviates from the ground truth. Table S3 lists all of the test error with a mean of 38 meters. It also shows that our network performs better in summer than other seasons.

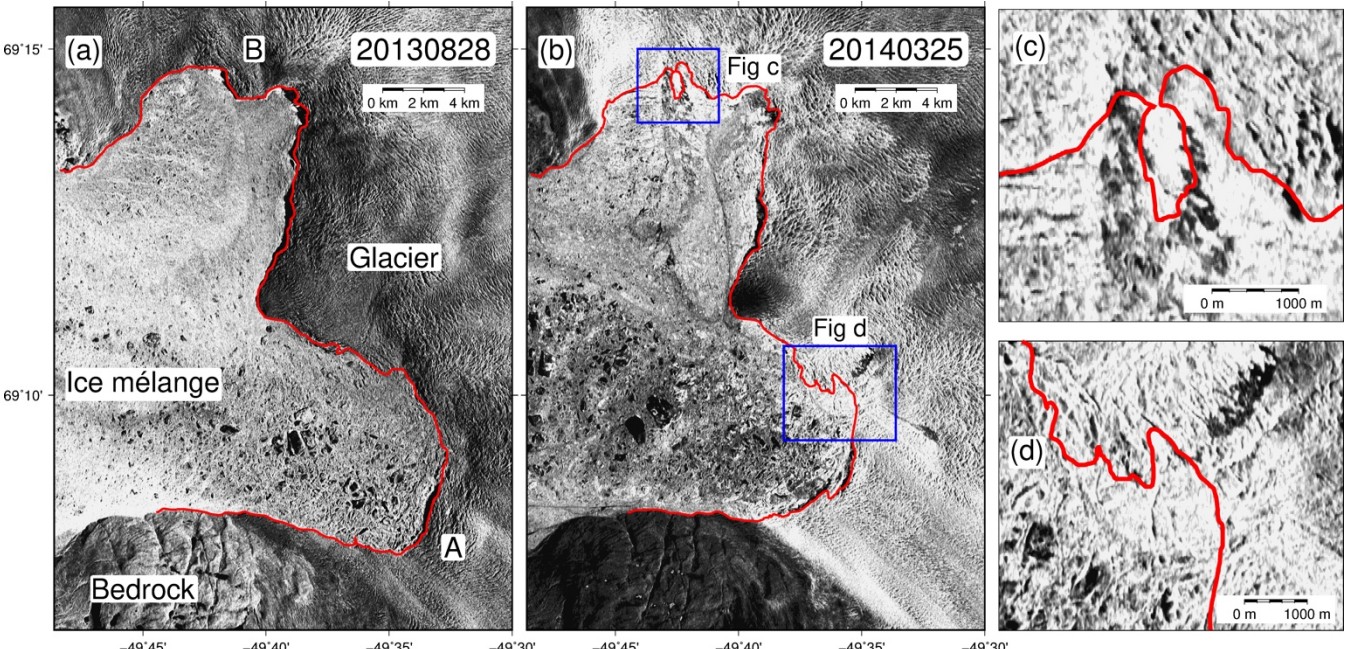

**Figure 4. Examples of (a) superior and (b) inferior delineation from our deep-learning-based method. In both (a) and (b), the red line shows the calving front delineated by the network. (c) and (d) show the zoom-in figure of the obscure calving front positions within the blue boxes in (b).**

Overall, our results agree well with the CCI products (Fig. 5). The area difference is $2.14 \times 10^6$ m$^2$, and the equivalent length

5   difference is 73 meters. Moreover, our results have a higher temporal resolution (about two measurements every month) than the CCI products (about four measurements every year). Therefore, we can observe the seasonal and inter-annual variations more clearly. Based on our results, Branches A and B retreated from 2009 to 2015 with linear trends of $-117 \pm 1$ m yr$^{-1}$ and $-157 \pm 1$ m yr$^{-1}$, respectively. The inter-annual variation can be roughly divided into three phases (Fig. 5 and summarized in Table 1). (1) From April 2009 to January 2011, the retreat rates were $-141$ m yr$^{-1}$ and $-228$ m yr$^{-1}$ along Branches A and B,

10  respectively. (2) From January 2011 to January 2013, the glacier retreated 170% and 61% faster than in the previous phase in Branches A and B, respectively. (3) From January 2013 to December 2015, these two branches behaved differently. In Branch A, the glacier retreated and advanced seasonally, but at much slower average rates ($-23$ m yr$^{-1}$). In Branch B, the seasonal variations were minor, and the glacier retreated slowly ($-46$ m yr$^{-1}$).

**Table 1. Retreat rates in area and equivalent length during different phases.**

|  | Period | Mean retreat rate | |
|---|---|---|---|
|  |  | Branch A | Branch B |
| Area change | Apr 2009–Jan 2011 | $-3.07 \pm 0.05$ | $-4.97 \pm 0.09$ |
| ($10^6$ m$^2$ yr$^{-1}$) | Jan 2011–Jan 2013 | $-8.30 \pm 0.04$ | $-8.03 \pm 0.07$ |
|  | Jan 2013–Dec 2015 | $-0.50 \pm 0.03$ | $-1.01 \pm 0.03$ |
|  | Apr 2009–Dec 2015 | $-2.56 \pm 0.01$ | $-3.41 \pm 0.01$ |
| Equivalent length change (m yr$^{-1}$) | Apr 2009–Jan 2011 | $-141 \pm 4$ | $-228 \pm 9$ |
|  | Jan 2011–Jan 2013 | $-381 \pm 3$ | $-368 \pm 5$ |
|  | Jan 2013–Dec 2015 | $-23 \pm 2$ | $-46 \pm 2$ |
|  | Apr 2009–Dec 2015 | $-117 \pm 1$ | $-157 \pm 1$ |

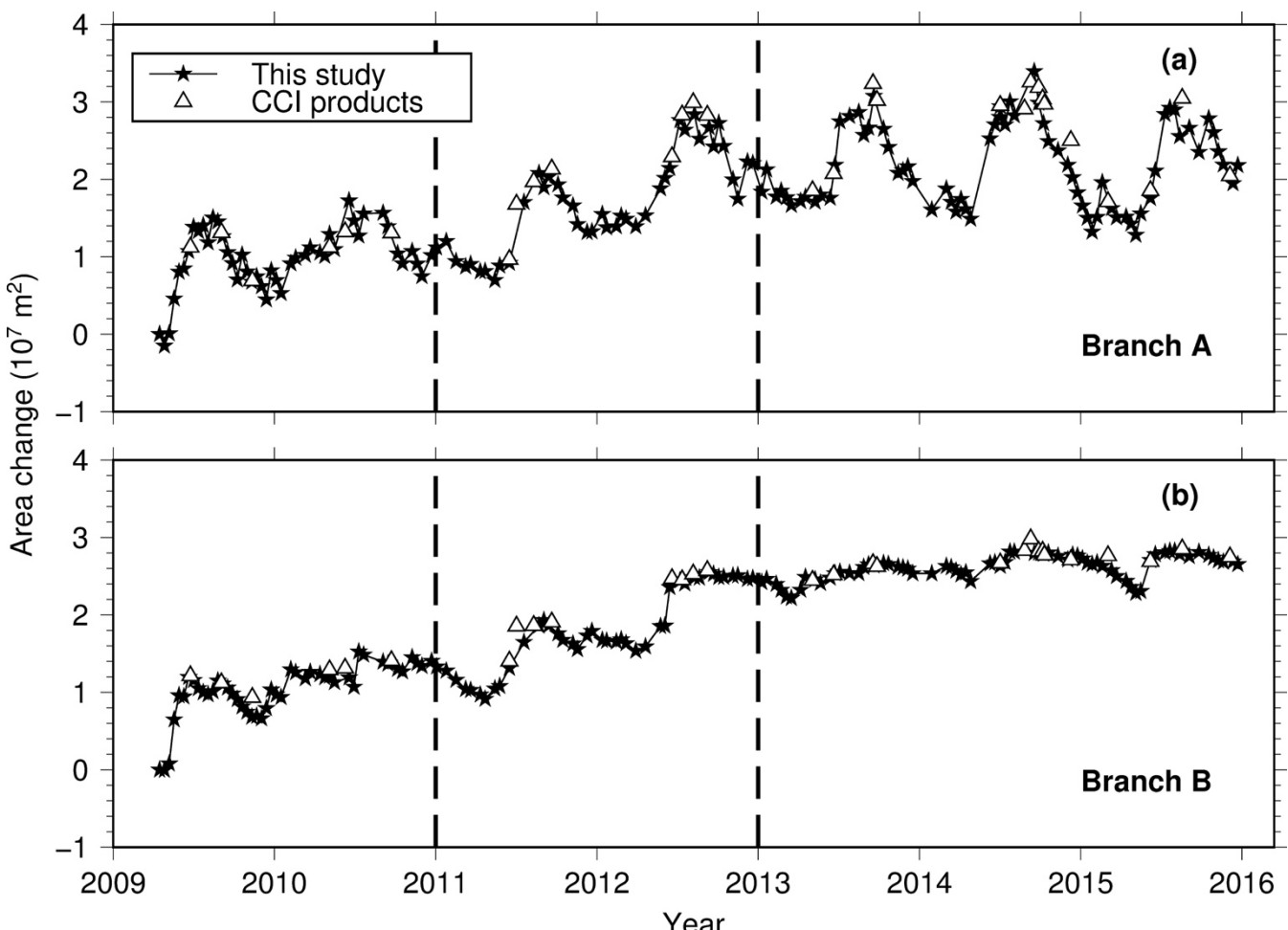

**Figure 5. Time series of calving front variations (in area changes) of Branches A and B from our deep learning method (stars) and the Greenland Ice Sheet CCI project (triangles). Dashed vertical lines divide the time series into three separate phases (see text).**

Further examination of the inter-annual variation indicates that the calving front exhibited different seasonal variations from
5   year to year. First, even within a close distance of ten kilometers around the coastal area, Branches A and B behaved asynchronously. For example, in 2010, Branch A began to retreat in May, while Branch B started to retreat one month later (Fig. 6a and 7). Moreover, after 2012, Branch A's front underwent strong seasonal variation while Branch B's front remained relatively stable (Fig. 6b). Second, the retreat timing of the glacier varied in different years. In Branch A, the front began to retreat around May in most years, while in 2011 and 2013 the retreat started in June. In 2010, both branches
10   experienced a sudden retreat from mid-January to early February, and then became stable. Third, the calving front variation became regular after 2012. In Branch A, the front stopped retreating in July of each year, and its position remained unchanged up to September to October. In Branch B, the front advanced in spring and retreated in early summer, while its position remained almost unchanged in other seasons (Fig. 5 and 7).

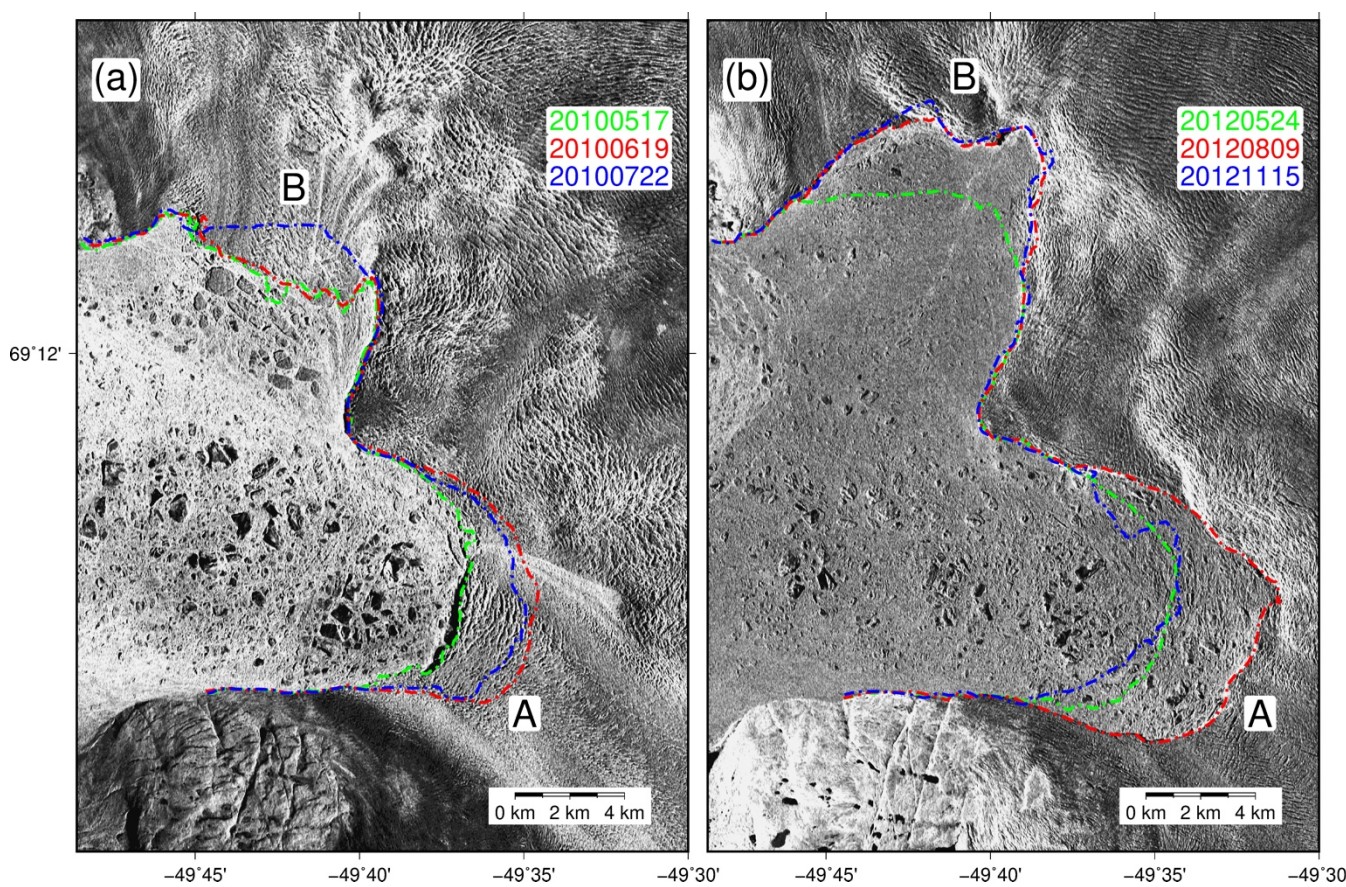

**Figure 6.** Two examples showing the asynchronous behaviors of Branches A and B. (a) Branch A began to retreat in May 2010, while Branch B started to retreat one month later. (b) Branch A's calving front underwent strong variation between August to November 2012, whereas Branch B's calving front was relatively stable. The magenta line in both (a) and (b) shows the calving front position just before the annual retreat.

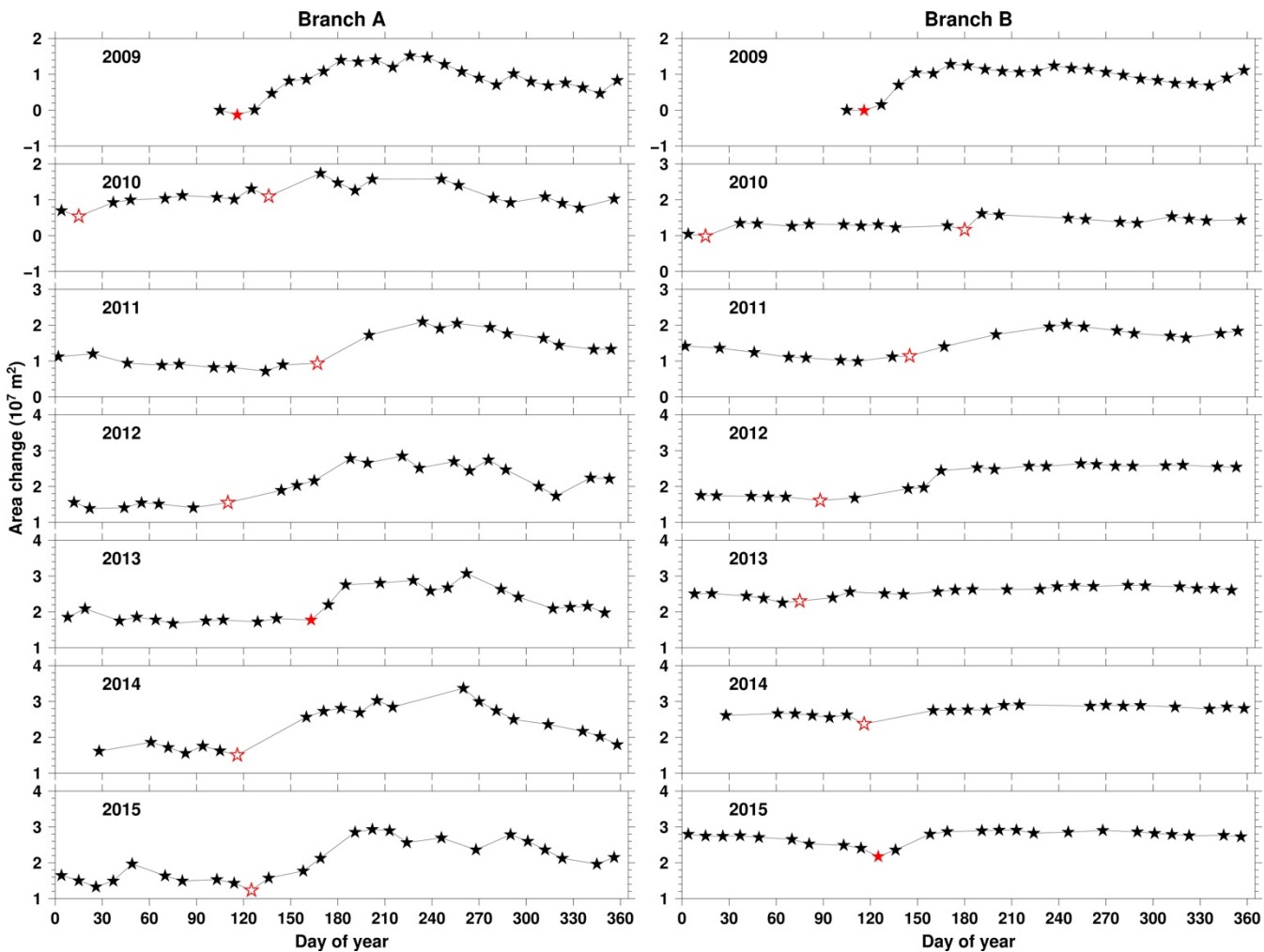

**Figure 7. Similar to Fig. 5 but showing the time series of calving front changes (in area changes) of our deep-learning-based results in different years. The red filled stars mark the dates when the glacier started to retreat. Red open stars mark starting dates that cannot be reliably determined due to data gaps (e.g., Branch A in 2011) and small variations (e.g., Branch B in 2013).**

5 ## 7 Discussion

### 7.1 Differences from the previous work

Mohajerani et al. (2019) have applied U-Net architecture to Landsat images over Jakobshavn, Sverdrup, Kangerlussuaq, and Helheim glaciers in Greenland. Despite both using the U-Net architecture, our study is different from Mohajerani et al. (2019) in datasets, result accuracy, transferability, strategies for classification, post-processing, and image resampling. The

10 usage of high-resolution TSX images allows us to generate more accurate calving fronts. Without additional manual practices, our method is more transferable, particularly when applying to large areas with many glaciers. Below we discuss the technical differences in detail.

First, our study classifies the surface into two types (i.e., ice mélange and non-ice mélange) to extract the calving front, while Mohajerani et al. (2019) used semantic segmentation to extract the front without classifying the surrounding surfaces. Both strategies require post-processing procedures. In our method, erroneous segmentation can cause small isolated polygons within the ice mélange or the non-ice mélange regions. Yet, we can solve this problem by removing these small polygons in the post-processing. The semantic segmentation used by Mohajerani et al. (2019) can be affected by icebergs, crevasses, etc. Nonetheless, the least-cost path search method could solve this problem (Mohajerani et al., 2019). Second, additional manual practices are needed in the work of Mohajerani et al. (2019). For instance, images of every single glacier in their work were adjusted by a certain angle to make all the glaciers flow in the same direction in the pre-processing. Third, we subdivide the images into small patches, which allows us to utilize the advantages of images with high resolution and various sizes. Mohajerani et al. (2019) resampled images to a fixed size (240 by 152 pixels) with low spatial resolution (49.0 to 88.1 meters), therefore the position accuracy is limited.

## 7.2 Calving front variation and bed elevation

In general, calving front variations are influenced by multiple factors, including floating or grounding conditions (McFadden et al., 2011; Murray et al., 2015; Bondzio et al., 2017; Fried et al., 2018), interaction with the ocean (Holland et al., 2008; Howat et al., 2008; Motyka et al., 2011; Vieli and Nick, 2011; Straneo et al., 2013), ice mélange and sea ice conditions (Amundson et al., 2010; Moon et al., 2015; Cassotto et al., 2015), basal lubrication (Joughin et al., 2008b; Moon et al., 2014) and bed elevation (Joughin et al., 2008a; Joughin et al., 2014; Kehrl et al., 2017; Bunce et al., 2018). Here we examine the possible link between the observed variations of the calving fronts with bed elevation.

Bed elevation has a substantial influence on the glacier retreat. In the first situation where the bed is flat, glacier retreat decreases resisting force, which accelerates the glacier. The acceleration of the glacier can also thin the ice. Thinning reduces the effective pressure at the bed, $N = P_i - P_w$, where $P_i$ is the overburden pressure and $P_w$ is water pressure. A decreased $N$ reduces basal drag, causing stretching and faster flow and constituting positive feedback. In the second situation, as the glacier retreats into an overdeepened basin where the bed slopes down inland or is retrograde, the positive feedback is reinforced, and the glacier becomes more unstable, for two reasons. First, ice thickness at the calving front increases as the retreat progresses, increasing driving stress. Second, because the calving front moves into deeper water, this retreat decreases $N$ further. In the third situation where the bed slopes inland-uphill, the glacier may stabilize, since retreating into shallower water increases $N$ and decreases driving stress.

Previous studies also suggest that bed elevation has a substantial influence on glacier calving front variations. Examining the height above flotation of Branch A in Jakobshavn, Joughin et al. (2014) suggested that retreating into an overdeepened basin where the bed slope is retrograde may lead to an unstable calving front retreat, and a bed sloping inland-uphill may stabilize the glacier. Other studies have also suggested that retreating into deeper water may accelerate the glacier, resulting in an unstable retreating (Howat et al., 2005; Howat et al., 2007; Nick et al., 2009; Catania et al., 2018).

In our study area, the bed elevation derived from BedMachine v3 (Morlighem et al., 2017) shows two overdeepened basins along the main channel of Branch A (Fig. 8a). During the period from 2009 to 2015, the calving front of Branch A retreated into the second overdeepened basin in August 2011 for the first time, which may have produced a faster rate of retreat. In July 2012, the glacier retreated to the bottom of the overdeepened basin and stopped retreating further (Movie S2). The

inland-uphill bed slope behind the bottom of the overdeepened basin may have prevented the glacier from further retreating. In Branch B, after June 2012, the glacier retreated into a zone where the bed slopes uphill inland (Fig. 8b, Movie S3). We suggest that retreating into this zone may have led to the more regular and stable behavior of Branch B after June 2012 (Fig. 5b).

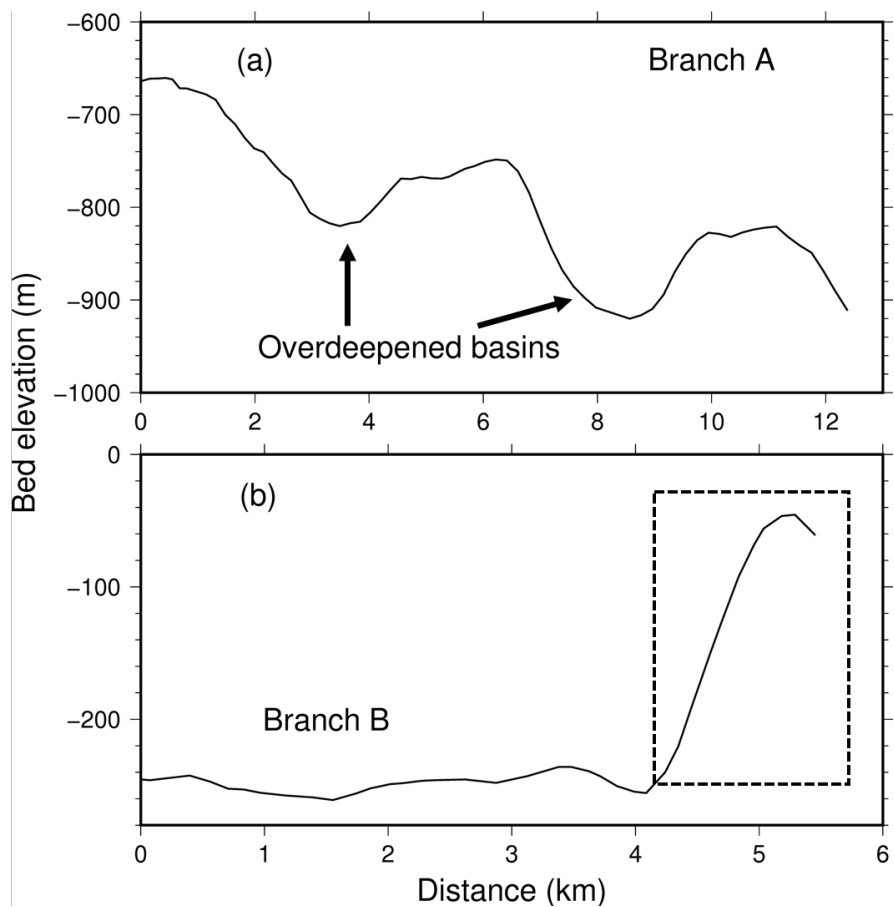

**Figure 8. Bed elevation profiles of two branches derived from BedMachine v3 (Morlighem et al., 2017). The profile locations are shown in Fig. 1a. The dashed box shows the zone where the bed slopes uphill inland.**

### 7.3 Limitations of current method

The current method is limited by high computational power requirement, and manual delineation largely control its accuracy. First, the U-Net architecture requires relatively high GPU memory for large images. In our configuration, around 15 gigabyte

(GB) GPU memory is needed for training the network. Second, although splitting images with overlaps allow as to apply the

network to images with different sizes, the overlaps increase the training time. These two limitations can be overcome by hardware development. With more powerful GPU in the future, we can increase the calculation efficiency and lessen the training time. Third, the accuracy of this method relies on manual delineation as well as the information richness of the training dataset (Goodfellow et al., 2016). If the training examples are not representative for the actual task or if the manual delineation in these examples is of low quality or inconsistent, U-Net will either fail to train or will reproduce inconsistent results on new data. To further increase the accuracy and robustness of the network, more training examples are needed.

## 7.4 Prospects for future work

In the near future, we will include more training examples to minimize network error. In this study, the well-trained network is limited to a specific dataset, namely TSX images. However, it is feasible to apply the DCNN to multi-sensor remote sensing imagery, which has been proved by previous studies (Nogueira et al. 2017; Lang et al., 2018). Moreover, as long as the calving fronts are clear in the images, our method can also use images with light cloud cover and Landsat 7 images with scan line errors.

The effectiveness and transferable nature of deep learning (Anantrasirichai et al., 2018) promises that our methodology can be applied to many other glaciers, both in Greenland and elsewhere in the world. Besides Jakobshavn Isbræ, other Greenland tidewater glaciers such as Helheim and Kangerdlugssuaq also show strong calving front variations (Howat et al., 2005; Howat et al., 2007; Joughin et al., 2008a). In theory, the DCNN can be retrained whenever new data is added to the training dataset. Moreover, including more data over other places can increase the generalization of the network, making it applicable to more situations (Goodfellow et al., 2016).

## 8 Conclusions

This study designs a method based on DCNNs to automatically delineate calving fronts of Jakobshavn Isbræ from TerraSAR-X SAR images acquired from April 2009 to December 2015. Small test error suggests that the accuracy of a well-trained network can be close to the human level. Our results reveal that the two branches of Jakobshavn Isbræ behaved asynchronously. We suggest that bed elevation may have a major influence on the observed calving front variations. Our methodology can be applied to many other tidewater glaciers both in Greenland and elsewhere in the world using multi-temporal and multi-sensor remote sensing imagery.

## Code and data availability

The Greenland Ice Sheet CCI products are available from http://products.esa-icesheets-cci.org. The Bed elevation (BedMachine v3) is available from http://sites.uci.edu/morlighem/dataproducts/bedmachine-greenland. The code of the

whole framework (Fig. 3) will be provided by Enze Zhang upon request. The network-delineated calving fronts obtained in this study are available from https://doi.pangaea.de/10.1594/PANGAEA.897064 (Zhang et al., 2019).

## Author contribution

EZ developed the code, performed the TSX images analysis and calving front quantification, and wrote the manuscript. LL advised EZ and edited the manuscript. LH actively helped to develop the code.

## Competing interest.

The authors declare that they have no conflict of interest.

## Acknowledgments

We acknowledge the support of NVIDIA Corporation, which donated the donation of the Quadro P5000 GPU used for this study. The TerraSAR-X SAR images were provided by the Deutsches Zentrum für Luft-und Raumfahrt (DLR) (LAN1650, LAN1797, and COA3605). Our deep learning code is based on Jakeoung Koo's work (https://github.com/jakeoung/Unet_pytorch). This study was supported by the Hong Kong Research Grants Council (CUHK24300414) and CUHK Direct Grant for Research (4053282). We highly appreciate the anonymous reviewers for their constructive comments and suggestions, which significantly improve the quality of this paper.

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
