# Peer review of "Automatically delineating the calving front of Jakobshavn Isbræ from multi-temporal TerraSAR-X images: a deep learning approach"

_The Cryosphere, 2019_

## Referee Comment (RC1) · Anonymous Referee #1 · 11 Mar 2019

General comments:

This paper presents the application and study of emerging machine learning techniques towards automatic calving front detection. Specifically, it utilizes a deep neural network architecture, U-Net, to automatically segment raw SAR imagery along calving fronts into digitized vectors. This study focuses on Jakobshavn from 2009-2015, and performs analysis using additional data products to cross-validate the results. The analysis correlates and validates data from the Greenland Ice Sheet CCI project, Bed-Machine v3 bedrock data, and the automatically determined calving fronts from this paper. Images to describe the study, and accompanying data tables, help communi-

cate the work done.

The paper is well written and covers a novel emerging technique (deep learning in the cryosphere). Therefore, I would like to recommend it for publication. However, I do have two concerns, though it may not be within the scope of this paper. These concerns regard the paper's wider implications/context, and may impact the rigor/novelty/impact of this study.

The first concern relates to existing similar work conducted by Mohajerani, Y., et. al., in Remote Sensing. Please refer to their paper here: https://www.mdpi.com/2072-4292/11/1/74/htm. While the methods are no doubt similar (deep-learning UNet), the one covered in this paper seems to be more accurate and more comprehensively analysed, though by virtue of being more focused in scope. For comparison, this paper covers Jakobshavn, TerraSAR-X, while Mohajerani covers Jakobshavn, Sverdrup, Kangerlussuaq, Helheim, Landsat 8 in Mohajerani's paper. I think it is helpful to have corroborating evidence of the validity of this methodology - especially published in The Cryosphere. Regardless, while I can still make my recommendation, I will leave others to discuss this matter.

The second concern I have relates to the generalizability of the network. While I acknowledge this is not the focus of the case study, the following are some questions I, and perhaps others, would express interest in knowing.

Specific Comments

Page 7 Line 1 - It was mentioned that summer imagery has higher performance than winter imagery. Though the ice melange has similar texture to glacial ice, should it not be possible for further training to be performed to close this gap? Perhaps the network needs additional capacity to handle this differentiation?

Page 15 Line 6-8 - It is mentioned that this methodology can be applied to other domains. Do you have any analyses on how the network performs on other glacial domains, such as Sverdrup, or Helheim?

Page 2 Line 29 - Does this network rely on features only visible at 3.3-3.5m? i.e., does lowering the pixel resolution adversely affect accuracy/performance? -Similarly, can the network handle lower resolution 30/60m datasets like Landsat?

Page 2 Line 34 - It is mentioned that the cloud cover issue is avoided. However, some light cloud cover does not always obscure calving front edges. Would it be feasible to train the network to handle these issues, to allow greater temporal resolution/constraints by not eliminating minor cloud covered images from the study? By extension, could the network handle Landsat 7 scan line errors, given additional training?

If this application/methodology is to be impactful beyond the scope of this study, it must be able to perform well on other not just on other conditions, but on other glacial domains, or even other datasets, without too much additional effort in retraining. Again however, it is not necessary to cover this, as I understand the above is not within the scope of the paper.

Nevertheless, this paper still represents a good case study on the application of these emerging machine learning techniques as applied to the cryosphere.

---

## Referee Comment (RC2) · Anonymous Referee #2 · 20 Mar 2019

The authors use a deep convolutional neural network with a U-net architecture to delineate the calving fronts of Jakobshavn Isbrae between 2009 and 2015. The network achieves reasonable results, allowing the analysis of the interannual and season behavior of the two branches of the glacier. The authors determine three distinct phases of calving front behavior, which they partially attribute to the bed elevation. There are a some issues with the manuscript regarding originality of the paper, ambiguous or incorrect technical comments, and lack of clarity in some aspects of the methods. However, it does add valuable results and showcases the uses of deep learning in SAR products. Therefore, I believe the article may be considered for publication after Major Revisions, once the following concerns have been addressed:

[Figure]

MAJOR COMMENTS:

1. As the first reviewer pointed out, despite the claim in the manuscript regarding the novelty of the technique, the methodology is very similar to that of Mohajerani et al. [2019] (https://doi.org/10.3390/rs11010074). However, this study does provide a different take on this technique and the authors should point out specifically how this work improves on previous efforts. For instance, the authors here use classification of surfaces in order to obtain the calving front, while Mohajerani et al use semantic segmentation to extract the front without classifying the surrounding surfaces. Each technique has strengths in different contexts. This and other differences should be discussed.

2. There are some statements that are not necessarily true from a technical point of view and raise some concern, which require revision:

i) Page 6 Lines 12-15: This is not true. Even when using one architecture, the loss and/or accuracy metrics on the validation dataset can be used during training in order to avoid overfitting, whereas the test dataset is only used after training. This is particularly important if the trained network is intended to be used in multiple areas.

ii) Page 7 Lines 7-8: This statement is not necessarily true and could be misleading. A larger kernel provides more context, but doesn't necessarily directly increase precision. It is dependent on the scale of the desired features to be extracted, depth of network, desired level of weight sharing, and many other factors.

iii) Page 7 Line 27: It is not necessarily true that having more items in a batch reduces overfitting. This is dependent on the total number of epochs that the batches are cycled through and the rate of minimization of the loss function as a function of batch size. Large batches can indeed reduce generalizability (e.g. Keskar et al [2016] https://arxiv.org/abs/1609.04836).

3. There is no proper measure of the extent of overfitting in the study. Without a

validation dataset to keep track of overfitting during training, and no regularization in the network (or lack of discussion in the manuscript), one cannot make any statements about the generalizability of the model. This is exacerbated by the fact that the authors train and test the network on only one and the same glacier.

4. It would be helpful to provide more detailed information on the time requirements (e.g. Page 7 Lines 30-31) and the GPU model used in the study as a point of reference.

5. There is very little discussion on the actual architecture of the U-Net model. How many layers are used, what activation functions are used, etc.?

6. It would be more meaningful to put the errors in context. For example Page 8 Line 28, how much of the error is purely from the delineation alone, if you had multiple investigators manually delineate the same calving front? And how do these errors and those reported in Table S3 compare with the resolution of the image in terms of the number of pixels?

MINOR COMMENTS:

Page 1 Line 16: add "to" after "stabilized".

Page 3 Line 13: change "speeded up" to "sped up"

Table S1: please statement more clearly if 0=test and 1=train to avoid confusion.

Page 4 Line 15: How are boundaries dealt with in the averaging of pixels?

Page 7 Lines 3-4: It is not very clear how the calving front is delineated front the closest temporal neighbor. Is there a set distance threshold from the calving front of the reference image?

Figure S4: "(c) and (c) show the manually delineated calving fronts" should be changed to "(c) and (d) [. . .]".

Page 7 Line 19: Is rotation augmentation necessary if you are only working with one

glacier here?

Page 7 Line 20: Please explain what you mean by 2% linear stretch. Is this done separately in each direction (horizontal and vertical)?

Page 8 Lines 3-4: Just a suggestion: in order to avoid losing training data, you can change the weights in the loss function instead.

Page 8 Line 9: what threshold do you use to determine a "stable error"?

Figure 10: the magenta and red colors are very hard to distinguish. Please consider using a more contrasting color.

Section 7.2: What are the limitations of the current technique? Could imagery artifacts or more varied surfaces be dealt with? Can the trained network be applied to multiple glaciers or does it have to be retrained for every glacier?

---

## Author Comment (AC1) · 30 Apr 2019

**General Comments**

This paper presents the application and study of emerging machine learning techniques towards automatic calving front detection. Specifically, it utilizes a deep neural network architecture, U-Net, to automatically segment raw SAR imagery along calving fronts into digitized vectors. This study focuses on Jakobshavn from 2009-2015, and performs analysis using additional data products to cross-validate the results. The analysis correlates and validates data from the Greenland Ice Sheet CCI project, Bed Machine v3 bedrock data, and the automatically determined calving fronts from this paper. Images to describe the study, and accompanying data tables, help communicate the work done.

The paper is well written and covers a novel emerging technique (deep learning in the cryosphere). Therefore, I would like to recommend it for publication. However, I do have two concerns, though it may not be within the scope of this paper. These concerns regard the paper's wider implications/context, and may impact the rigor/novelty/impact of this study.

We highly appreciate the reviewer for the constructive comments which have significantly improved the quality of our manuscript. We have made our best effort to revise the manuscript based on the referee's comments and suggestions.

The first concern relates to existing similar work conducted by Mohajerani, Y., et. al., in Remote Sensing. Please refer to their paper here: https://www.mdpi.com/2072-4292/11/1/74/htm. While the methods are no doubt similar (deep-learning UNet), the one covered in this paper seems to be more accurate and more comprehensively analysed, though by virtue of being more focused in scope. For comparison, this paper covers Jakobshavn, TerraSAR-X, while Mohajerani covers Jakobshavn, Sverdrup, Kangerlussuaq, Helheim, Landsat 8 in Mohajerani's paper. I think it is helpful to have corroborating evidence of the validity of this methodology - especially published in The Cryosphere. Regardless, while I can still make my recommendation, I will leave others to discuss this matter.

We have added a new subsection 7.1 titled **Differences from the previous work** to discuss the differences between our work and the method of Mohajerani et al. (2019), which are summarized as follows:

- Different strategies are used to classify calving fronts. Our study classifies the surface into two types (i.e., ice mélange and non-ice mélange) to extract the calving front; Mohajerani et al. (2019) use semantic segmentation to extract the front without classifying the surrounding surfaces.
- Additional manual practices such as finding a rotation angle for each glacier are needed in the work of Mohajerani et al. (2019).
- We subdivide the images into small patches, which allows us to use images with high resolutions and various size (i.e., TerraSAR-X images). Mohajerani et al. (2019) resampled images to a fixed size (240 by 152 pixels) with low spatial resolution (49.0 to 88.1 meters).

The second concern I have relates to the generalizability of the network. While I acknowledge this is not the focus of the case study, the following are some questions I, and perhaps others, would express interest in knowing.

The generalizability of the network relies on the diversity of the training examples. With additional training examples, our method can be applied to other places using multi-sensor remote sensing datasets. Moreover, optical images with low cloud cover and Landsat 7 images with scan line errors can be used as long as the calving fronts are visually clear. See our replies to the specific comments below for more details. We did not include the results at another other domains or the results using other remote sensing datasets since they are preliminary and beyond the scope of this manuscript.

**Specific Comments**

Page 7 Line 1 - It was mentioned that summer imagery has higher performance than winter imagery. Though the ice melange has similar texture to glacial ice, should it not be possible for further training to be performed to close this gap? Perhaps the network needs additional capacity to handle this differentiation?

It is possible to close this gap by including more winter training examples. The accuracy of the well-trained network relies on the quality of the training examples. Delineating calving fronts in winter images with blur boundaries is challenging, and therefore the quality of winter training examples is not as good as those in summers. Including more winter training examples could make the trained network more robust and therefore mitigate the problem caused by winter data quality. However, due to the quota limitation, we only have 159 TerraSAR-X images. Therefore, we did not close this gap in the current work. Note that we did not include the discussion about the possibility to close the gap since it is beyond the scope of this manuscript.

Page 15 Line 6-8 - It is mentioned that this methodology can be applied to other domains. Do you have any analyses on how the network performs on other glacial domains, such as Sverdrup, or Helheim?

We conducted a preliminary experiment by directly applying the network generated from this work as trained by TerraSAR-X imagery from Jakobshavn to Helheim (without including any new training data). Figure R1 shows that the automatically delineated calving front at Helheim is very close to what one would get from visual inspection. Therefore, our method can be applied to other glaciers. Of course, we need to include more training examples from more glaciers to ensure reliable results on other glacier domains.

Figure R1. An example of automatically delineated calving front at Helheim. The background image is a Landsat 8 image taken on April 11th, 2015. The red line indicates the automatically delineated calving front.

Page 2 Line 29 - Does this network rely on features only visible at 3.3-3.5m? i.e., does lowering the pixel resolution adversely affect accuracy/performance? -Similarly, can the network handle lower resolution 30/60m datasets like Landsat?

This network does not rely on high-resolution images. As long as the calving front is visually clear, the network is able to handle images with different resolutions and sizes. For example, with additional training, the network can generate reasonable results using lower resolution image such as Landsat, as shown in Figure R2. Note that training dataset used to train this network does not include the image in Figure R2.

---

## Author Comment (AC2) · 30 Apr 2019

The authors use a deep convolutional neural network with a U-net architecture to delineate the calving fronts of Jakobshavn Isbrae between 2009 and 2015. The network achieves reasonable results, allowing the analysis of the interannual and season behavior of the two branches of the glacier. The authors determine three distinct phases of calving front behavior, which they partially attribute to the bed elevation. There are a some issues with the manuscript regarding originality of the paper, ambiguous or incorrect technical comments, and lack of clarity in some aspects of the methods. However, it does add valuable results and showcases the uses of deep learning in SAR products. Therefore, I believe the article may be considered for publication after Major Revisions, once the following concerns have been addressed:

We highly appreciate the reviewer for the constructive comments which have significantly improved the quality of our manuscript. We have made our best effort to revise the manuscript based on the referee's comments and suggestions.

**Major Comment**

As the first reviewer pointed out, despite the claim in the manuscript regarding the novelty of the technique, the methodology is very similar to that of Mohajerani et al. [2019] (https://doi.org/10.3390/rs11010074). However, this study does provide a different take on this technique and the authors should point out specifically how this work improves on previous efforts. For instance, the authors here use classification of surfaces in order to obtain the calving front, while Mohajerani et al use semantic segmentation to extract the front without classifying the surrounding surfaces. Each technique has strengths in different contexts. This and other differences should be discussed.

We have added a new subsection 7.1 titled **Differences from the previous work** to discuss the differences between our work and the method of Mohajerani et al. (2019), which are summarized as follows:

- Different strategies are used to classify calving fronts. Our study classifies the surface into two types (i.e., ice mélange and non-ice mélange) to extract the calving front; Mohajerani et al. (2019) use semantic segmentation to extract the front without classifying the surrounding surfaces.
- Additional manual practices such as finding a rotation angle for each glacier are needed in the work of Mohajerani et al. (2019).
- We subdivide the images into small patches, which allows us to use images with high resolutions and various size (i.e., TerraSAR-X images). Mohajerani et al. (2019) resampled images to a fixed size (240 by 152 pixels) with low spatial resolution (49.0 to 88.1 meters).

There are some statements that are not necessarily true from a technical point of view and raise some concern, which require revision:

i) Page 6 Lines 12-15: This is not true. Even when using one architecture, the loss and/or accuracy metrics on the validation dataset can be used during training in order to avoid

overfitting, whereas the test dataset is only used after training. This is particularly important if the trained network is intended to be used in multiple areas.

> We agree and have separated our data into three parts: training, validation, and test. We revised the relevant text as: *We separate all the SAR images into a training-validation dataset (75 images) and a test dataset (84 images) (Table S1). In the training-validation dataset, we randomly choose 90% as training data and take the rest as validation data.* (Page 7 Line 13-15)

ii) Page 7 Lines 7-8: This statement is not necessarily true and could be misleading. A larger kernel provides more context, but doesn't necessarily directly increase precision. It is dependent on the scale of the desired features to be extracted, depth of network, desired level of weight sharing, and many other factors.

> Indeed, the accuracy relies on several factors such as the depth of the network and desired level of weight sharing. The primary purpose of increasing the kernel size is to get smoother calving fronts. We rephrased the relevant text as: *We utilize relatively large convolution kernel size (5 by 5) to obtain smoother calving fronts.* (Page 7 Line 7)

iii) Page 7 Line 27: It is not necessarily true that having more items in a batch reduces overfitting. This is dependent on the total number of epochs that the batches are cycled through and the rate of minimization of the loss function as a function of batch size. Large batches can indeed reduce generalizability (e.g. Keskar et al [2016] https://arxiv.org/abs/1609.04836).

> We agree that a larger batch size would not reduce overfitting but actually reduce generalizability. Typically, batch sizes are no larger than 256. A large batch size would help to increases the efficiency and improves the accuracy of the gradient estimation at each step. Here, the batch size we use is three. We revised the relevant text as: *With a given GPU memory, a smaller patch size allows more items in a batch, which increases the efficiency and improves the accuracy of the gradient estimation at each step. To strike a balance between edge effect and batch size, we choose 960×720 pixels as our patch size and the batch size is three.* (Page 8 Line 22-24)

There is no proper measure of the extent of overfitting in the study. Without a validation dataset to keep track of overfitting during training, and no regularization in the network (or lack of discussion in the manuscript), one cannot make any statements about the generalizability of the model. This is exacerbated by the fact that the authors train and test the network on only one and the same glacier.

We have added the validation dataset and halted the training when the validation error stops to decrease with patience of 5 epochs (Page 7 Line 13-15; Page 9 Line 1-2). The optimizer we use has an L2 regularization term with a factor of 0.00001 (Page 7 Line 12). These strategies help to mitigate overfitting. We chose not to include the dropout layer because we found that

adding a dropout layer caused large fluctuations for both the training loss and validation loss at the end of training.

It would be helpful to provide more detailed information on the time requirements (e.g. Page 7 Lines 30-31) and the GPU model used in the study as a point of reference.

We have provided more detailed information on the time requirements (Page 8 Line 27). We do mention the used GPU model, Quadro P5000 GPU, in the Acknowledgment section. We prefer not to mention any brand name in the main part.

There is very little discussion on the actual architecture of the U-Net model. How many layers are used, what activation functions are used, etc.?

We have added one paragraph and a graph to describe the U-Net architecture (Page 6 Line 13, Page 7 Line 1-12, Figure S1). The architecture we use has 41 layers in total, including 23 convolutional layers and 18 batch normalization layers. The activation function in the last convolutional layer is Sigmoid, and the rest activation functions are LeakyReLU.

It would be more meaningful to put the errors in context. For example Page 8 Line 28, how much of the error is purely from the delineation alone, if you had multiple investigators manually delineate the same calving front? And how do these errors and those reported in Table S3 compare with the resolution of the image in terms of the number of pixels?

We agree that including the error from delineation alone would be more meaningful. We asked another investigator to manually delineate the calving fronts from six selected images. By comparing the two sets of independent delineation results, we obtained a mean difference of 33 meters (equivalent to ~5.5 pixels). We revised the relevant text as: *To measure the manual delineation error, we have another investigator to manually delineate the above-mentioned six calving fronts again. By comparing the two sets of independent delineation results, we obtained a mean difference of 33 meters (equivalent to ~5.5 pixels) (Table S2).* (Page 9 Line 23-26)

We have added the error in terms of the number of pixels in Table S3.

**Minor Comments**

Page 1 Line 16: add "to" after "stabilized".

We have revised as suggested (Page 1 Line16).

Page 3 Line 13: change "speeded up" to "sped up"

We have revised as suggested (Page 1 Line15).

Table S1: please statement more clearly if 0=test and 1=train to avoid confusion.

We have revised the caption of Table S1 as suggested.

Page 4 Line 15: How are boundaries dealt with in the averaging of pixels?

The images we use to delineate the calving front manually and to apply to the network are all multi-looked images. The original TerraSAR-X images have a high spatial resolution, and their pixel size is 1.25 meters. After reducing the image size by 25 times, the boundaries in the multi-looked images remain visually clear.

Page 7 Lines 3-4: It is not very clear how the calving front is delineated front the closest temporal neighbor. Is there a set distance threshold from the calving front of the reference image?

If the boundary is not clear in an image, we will find its closest temporal neighbor with a clear edge. By observing the texture variation due to the glacier movement, we can approximately decide where the calving front is for the blur image. Figure S3 gives an example of how we dealt with this issue. The manual delineation is all based on visual observations without any quantitative analysis.

Figure S4: "(c) and (c) show the manually delineated calving fronts" should be changed to "(c) and (d) [. . .]".

We have revised the caption of Figure S3.

We have changed the order of the Figures in supporting information in the order they are referred to in the main manuscript:

Figure S1--> Figure S2

Figure S2--> Figure S4

Figure S3--> Figure S5

Figure S4--> Figure S3.

We have added one figure in supporting information to describe the network architecture (Figure S1).

Page 7 Line 19: Is rotation augmentation necessary if you are only working with one glacier here?

Without rotation augmentation, the trained network still can generate reasonable results. However, we prefer to keep the rotation augmentation since it could be helpful when we apply our method to other glaciers in the future.

Page 7 Line 20: Please explain what you mean by 2% linear stretch. Is this done separately in each direction (horizontal and vertical)?

We didn't do the linear stretch separately in each direction.

The linear stretching is to change the pixels' values to increase the contrast.

For all values between 2% and 98% of the pixel value range, we use the following equation to do the linear stretching

$$P_{stretched} = 255 * \frac{(P_{in} - P_{min})}{(P_{max} - P_{min})}.$$

Where $P_{stretched}$ is the pixels' value after linear stretching and $P_{in}$ is the pixels' value before stretching. $P_{min}$ and $P_{max}$ are the 2nd and 98th percentile in the histogram (that is, 2% of the pixels have values lower than $P_{min}$, and 2% of the pixels have values larger than $P_{max}$ ).

For values lower than $P_{min}$, they are set as zero, and for values larger than $P_{max}$, they are set as 255.

We believe that "x% linear stretch" is a widely used terminology in remote sensing and therefore choose not to provide a detailed explanation in the manuscript.

Page 8 Lines 3-4: Just a suggestion: in order to avoid losing training data, you can change the weights in the loss function instead.

Thanks for your suggestion, but we prefer dropping out these one-class patches. By changing the weights in the loss function, we can indeed avoid losing training data. However, the primary purpose of dropping out one-class patches is to save computational power. The network may generate erroneous segmentation in the region that is away from the calving fronts due to dropping out one-class patches. However, we can fix this problem in post-processing by removing small isolated polygons caused by erroneous segmentation.

Page 8 Line 9: what threshold do you use to determine a "stable error"?

We have changed our strategy to ovoid overfitting. With give patience of 5 epochs, if the validation loss stops to decrease, we halt the training process (Page 9 Line 1-2).

Figure 10: the magenta and red colors are very hard to distinguish. Please consider using a more contrasting color.

We have changed the line color from magenta to green.

Section 7.2: What are the limitations of the current technique?

We have added a new subsection 7.3 titled **Limitation of current method** to discuss the limitations of the current technique, which are summarized as follows:

- The U-Net architecture requires relatively high GPU memory.
- Splitting images with overlaps increase the training time.
- The accuracy of this method relies on manual delineation and the information richness of the training dataset.

Could imagery artifacts or more varied surfaces be dealt with?

As long as the calving fronts are clear in the images, imagery artifacts or more varied surfaces will not be a problem to the network. For example, with additional training, the network could handle images with low cloud cover (Figure R1) as well as Landsat 7 images with scan line errors (Figure R2). Note that the image in Figure R1 is not in the training dataset, and the image in Figure R2 is in the training dataset.

However, imagery artifacts such as image distortion need to be corrected by pre-processing procedures other than deep learning.

We did not include the results using Landsat -7 and -8 images since they are preliminary and beyond the scope of this manuscript.

[Figure]

Figure R1. An example of automatically delineated calving front at Jakobshavn Isbrae using a Landsat 8 image with clouds. The image was taken on August 27[th], 2014. The blue box indicates an area with low cloud cover.

[Figure]

Figure R2. An example of automatically delineated calving front at Jakobshavn Isbrae using a Landsat 7 image with scan line errors. The image was taken on July 24th, 2013.

Can the trained network be applied to multiple glaciers or does it have to be retrained for every glacier?

Currently, if we want to apply the network to other glaciers, retraining is needed. We conducted a preliminary experiment by directly applying the network generated from this work as trained by TerraSAR-X imagery from Jakobshavn to Helheim (without including any new training data). Figure R3 is a superior example shows that the automatically delineated calving front at Helheim is very close to what one would get from visual inspection. Of course, we need to include more training examples from more glaciers to ensure reliable results on other glacier domains.

However, with more and more data from different glaciers included in the training dataset, the trained network has the potential to be applied to another glacier without retraining.

We did not include the results on other glacier domains since they are preliminary and beyond the scope of this manuscript.

[Figure]

Figure R3. An example of automatically delineated calving front at Helheim. The background image is a Landsat 8 image taken on April 11[th], 2015. The red line indicates the automatically delineated calving front.

---

## Author Response (AR2)

**Reply to Minor Comments**

We thank the reviewers for their constructive comments and suggestions, which have significantly improved this paper. We have made our best effort to revise the manuscript based on the referee's comments and suggestions.

Page 8 Line 17: The authors explained in the response letter to reviewers that linear stretching is applied to the pixel values to increase the contrast. I think this needs to be explained in the main text as well, as this could be confused with a linear distortion to the dimensions of the image itself, which is also a common augmentation technique when training neural networks (to reduce sensitivity to specific geometric configurations).

We agree that explanation of linear stretching is needed. We revised the relevant text as: *Second, we apply 2% linear stretch to the training images to enhance the edges. For all the values between the 2nd and 98th percentiles of the pixel value histogram, we linearly stretch them to the range between 0 to 255. The values lower than the 2nd percentile are set to zero, and the values larger than the 98th percentile are set to 255.* (Page 8 Line 17-19)

Page 15 Lines 11-13: Convolutional neural networks have the intrinsic property of being applicable to consecutive small batches of an image instead of the whole image (due to the nature of convolution). Therefore, it is not true to say the architecture of previous studies need to be changed in order to be applicable to small batches rather than the whole image. The only difference is in post-processing.

We have deleted the relevant sentence:  (Page 15 Line 11)

Page 8 Line 27:change "takes longer time" to "takes more time"

We have revised as suggested. (Page 8 Line 29)

Page 9 Line 2: change "increase with patience of five epochs" to "increase for five consecutive epochs" (is it consecutive?)

Yes, it is consecutive. We have revised as suggested. (Page 9 Line 4)

[revised manuscript text omitted]